# Relationship between Some *Myostatin* Variants and Meat Production Related Calving, Weaning and Muscularity Traits in Charolais Cattle

**DOI:** 10.3390/ani13121895

**Published:** 2023-06-06

**Authors:** Tamás Csürhés, Ferenc Szabó, Gabriella Holló, Edit Mikó, Márton Török, Szabolcs Bene

**Affiliations:** 1National Association of Hungarian Charolais Cattle Breeders, 3525 Miskolc, Hungary; csurhest@yahoo.com (T.C.); info@charolais.hu (M.T.); 2Department of Animal Sciences, Albert Kázmér Faculty, Széchenyi István University, 9200 Mosonmagyaróvár, Hungary; 3Institute of Animal Husbandry Sciences, Kaposvár Campus, Hungarian University of Agriculture and Life Sciences, 7400 Kaposvár, Hungary; bene.szabolcs.albin@uni-mate.hu; 4Faculty of Agriculture, University of Szeged, 6800 Hódmezővásárhely, Hungary; miko.edit@mgk.u-szeged.hu

**Keywords:** *myostatin* alleles, *Q204X*, *F94L*, muscularity scores, calving and weaning traits

## Abstract

**Simple Summary:**

The objective of this study was to evaluate the effect of different *myostatin* alleles on muscularity of four body parts and overall muscularity, and, moreover, on calving ease, birth weight and 205-day weaning weight of weaned calves in the Hungarian Charolais population. Five *myostatin* alleles of 2046 calves were involved in the study. Among the *myostatin* alleles, the effect of *Q204X* was statistically proved (*p* < 0.01 and *p* < 0.05) on the 205-day weaning weight, muscle score of back, muscle score of thigh, loin thickness score and overall muscle development percentage. It would be advisable to pay more attention to this allele in the breeding program.

**Abstract:**

The slaughter value of live cattle can be assessed during visual conformation scoring, as well as by examining different molecular genetic information, e.g., the *myostatin* gene, which can be responsible for muscle development. In this study, the *F94L*, *Q204X*, *nt267*, *nt324* and *nt414* alleles of the *myostatin* gene (*MSTN*) were examined in relation to birth weight (BIW), calving ease (CAE), 205-day weaning weight (CWW), muscle score of shoulder (MSS), muscle score of back (MSB), muscle score of thigh (MST), roundness score of thigh (RST), loin thickness score (LTS), and overall muscle development percentage (OMP) of Charolais weaned calves in Hungary. Multi-trait analysis of variance (GLM) and weighted linear regression analysis were used to process the data. Calves carrying the *Q204X* allele in the heterozygous form achieved approximately 0.14 points higher MSB, MST and LTS, and 1.2% higher OMP, and gained 8.56 kg more CWW than their counterparts not carrying the allele (*p* < 0.05). As for the *F94L* allele, there was a difference of 4.08 kg in CWW of the heterozygous animals, but this difference could not be proved statistically. The other alleles had no significant effect on the evaluated traits.

## 1. Introduction

The value of slaughter animals, that is, the carcass composition and meat quality of meat-producing farm animals, such as slaughter cattle, can be reliably evaluated with post-slaughter muscle and fat measurements and laboratory tests. In beef production, however, slaughterhouse evaluation and laboratory meat quality testing are often impossible in the trade, as the animals are marketed on a live basis. Despite failing the mentioned objective evaluation possibilities, both the sellers and the buyers must be able to appraise, visually or in other ways, the meat production value of these animals.

The meat production value of slaughter animals can be evaluated with a high degree of accuracy based on several seen, measured and estimated conformation traits. A large number of research results from literary sources support the fact that the age, weight, sex, conformation, condition, muscle mass and shape of live animals provide reliable information about their meat production; however, some environmental factors can also play an important role [1]. The mentioned traits can be easily assessed by visual scoring. At the same time, some major genes or quantitative trait loci (QTL) have been identified related to meet quantity and quality [2,3]. The latter situation gives us the opportunity to perform tests on live animals, as DNA can be isolated from blood or other tissue, and the gene or gene variants affecting meat production can be detected. Such tests can be carried out early, before slaughter of animals at a young age.

An indicator of slaughter value could be *myostatin*, which is an extracellular cytokine mostly expressed in skeletal muscles and known to play a crucial role in the negative regulation of muscle mass [4,5].

Sellick et al. [6] studying the different variants of *MSTN* found that *F94L* was the only polymorphism consistently related to increased muscling. Wiener et al. [7] found that the *myostatin* allele with the 11-bp deletion (MH) segregating in the South Devon breed affected several traits related to beef production. The MH allele was associated with heavier calves at birth but slower growth, leading to lighter adult animals. Allais et al. [8] found the superiority of carcass traits of calves carrying one copy of the mutated allele (*Q204X* or *nt821*) over noncarrier animals was approximately +1 SD in the Charolais and Limousin breeds but was not significant in the Blonde d’Aquitaine. In the Charolais breed, for which the frequency was the greatest (7%), young bulls carrying the *Q204X* mutation presented a carcass with less fat, less intramuscular fat and collagen contents, and a clearer and more tender meat than those of homozygous-normal cattle. Hales et al. [9] reported that the average daily gain measured in Limousin heifers across the whole study (121 days) was greater with two copies of the *F94L* (homozygous) variant. According to Ceccobelli et al. [10], the heterozygous *MSTN* in Marchigiana bulls showed slight superiority in the carcass weight (heterozygote 426 kg and normal 405 kg) and meat quality parameters, although not always with statistical significance.

Looking at the relevant literature, even though there are many research results available on the effect of *myostatin* on meat production in cattle, especially in double-muscled cattle [11,12], relatively less is known about the effect of certain alleles in the Charolais breed. Based on previous data [13,14,15], it seems that there are significant differences between the phenotypic performance of individuals carrying and not carrying the *myostatin* alleles [16]. According to our opinion, this information is very important for improving performance, quality and genetic traits of the Hungarian Charolais population.

To our knowledge, phenotypic characteristics of calves related to *MSTN* alleles in the Hungarian Charolais population, even certain allele variants in the Charolais breed, has not been studied so far.

The objective of the present study was to evaluate some *myostatin* alleles such as *F94L* and *Q204X* and others (*nt267*, *nt324* and *nt414*) on birth weight, calving ease, 205-day weaning weight and muscle score of some body parts (shoulder, back, thigh and loin), and overall muscularity showing muscle development and trend of these traits in the Charolais beef cattle population in Hungary.

## 2. Materials and Methods

### 2.1. The Database

Data processed during the work were collected from the pedigree database, the National Association of Hungarian Charolais Cattle Breeders. The available and evaluated initial database contained pedigree, weaning, conformation traits and molecular genetic information. In the study, there were 2046 EU-registered weaned Charolais calves (688 male and 1358 female) born between 2015 and 2021.

### 2.2. The Studied Traits

During the study, the birth weight of calves (BIW), calving ease of dams (CAE), 205-day weaning weight of calves (CWW), muscle score of shoulder (MSS), muscle score of back (MSB), muscle score of thigh (MST), roundness score of thigh (RST), loin thickness score (LTS) and overall muscle development percentage (OMP) as phenotypic traits of weaned calves were evaluated in relation to *MSTN* mutations.

The conformation traits were scored at the weaning. The scoring of the mentioned body parts was carried out according to the Conformation Scoring Guideline of the National Association of Hungarian Charolais Cattle Breeders [17]. Each animal for each trait was scored from 1 to 10 points depending on the mass and shape of the muscles. However, the values of the OMP were calculated as the sum of the scores of each body part and the ratio of the maximum possible total score in per cent as follows:OMP = (MSS + MSB + MST + RST + 2 × LTS) / 60(1)

The calving ease of cows was scored as follows: normal light calving = 1, calving with assistance = 2 and difficult calving = 3.

### 2.3. The Molecular Genetic Informations

The molecular genetic information of the 2046 weaned calves was determined with the Weatherbys Scientific Bovine VersaSNP 50K chip. The description of the method and the possibilities of interpreting the results are described in detail by [18].

The genetic database contained information on 117 different alleles. In the course of this study, five relevant alleles of the gene encoding the *myostatin* protein (growth differentiation factor 8; GDF8), *F94L*, *Q204X*, *nt267*, *nt324* and *nt414* were examined [19,20]. Based on the available information [21,22,23], it seems that these alleles can have a significant impact on muscle growth, including the development of muscularity. In each case, it was indicated in the database whether the individuals carry the *F94L*, *Q204X*, *nt267*, *nt324* and *nt414* alleles in the homozygous or heterozygous form, or not. The distribution of these alleles by sex of calves is shown in Table 1.

### 2.4. The Effect of Different Factors

Before evaluating the database, the basic statistical parameters of the examined traits (mean, standard deviation, CV%, etc.) were calculated. The Kolmogorov–Smirnov test was used to check the normality of the data, and Levene’s test was used to check the homogeneity of the variances (Table 2).

To evaluate the database, the multifactor analysis of variance (general linear model) was applied [24]. During this work, the birth year and sex of the calves, as well as the genotype determined on the basis of the *myostatin* alleles (mentioned above), were incorporated into the model as fixed effects [16]. The nine examined traits were treated separately from each other, and in all nine cases separated models were performed. The general formula of the models used was as follows:ŷ_hijklmn_ = μ + Y_h_ + S_i_ + F_j_ + Q_k_ + N_l_ + M_m_ + T_n_ + e_hijklmn_(2)
where ŷ_hijklmn_ = trait of a weaned calf of “h” year, “i” sex, “j” *F94L*, “k” *Q204X*, “l” *nt267*, “m” *nt324* and “n” *nt414* genotypes; μ = average of all observations; Y_h_ = effect of birth year of calves; S_i_ = effect of sex of calves; F_j_ = effect of *F94L* allele; Q_k_ = effect of *Q204X* allele; N_l_ = effect of *nt324* allele; M_m_ = effect of *nt324* allele; T_n_ = effect of *nt414* allele; and e_hijklmn_ = random error [10].

### 2.5. Estimation of Phenotypic Trends and Phenotypic Correlations

For all nine traits, the data of the calves born in the same year were analyzed and averaged by year. Weighted one-way linear regression analysis was used to estimate the phenotypic trends. The dependent variable was the evaluated trait, the birth year of calves was considered as an independent variable, and the weight was the number of individuals per year.

Among the nine evaluated traits, Pearson’s phenotypic correlation values (r) were also determined.

### 2.6. The Used Softwares

The data were prepared using Microsoft Excel 2003 and Word 2003. The evaluation of the database was performed with the statistical software package SPSS 27.0 [25].

## 3. Results

For all traits, the influence of the sex and birth year of the calf was statistically verifiable (*p* < 0.01) and played a decisive role (62.27–96.74%) in the development of the phenotype (Table 3). The effect of the year of birth of the calves on the tested traits was also significant (*p* < 0.01). Among the *myostatin* alleles, the effect of *Q204X* was statistically proved (*p* < 0.01 and *p* < 0.05) on the traits CWW, MSB, MST, LTS and OMP. The other alleles had no effect on the evaluated weaning and muscularity traits.

The adjusted overall mean values (±SE) of the examined traits was as follows (Table 4 and Table 5): BIW 43.65 ± 0.63 kg, CAE 1.12 ± 0.05 points, CWW 269.07 ± 4.73 kg, MSS 5.90 ± 0.11 points, MSB 5.39 ± 0.11 points, MST 5.65 ± 0.12 points, RST 5.54 ± 0.12 points, LTS 5.52 ± 0.11 points and OMP 55.86 ± 0.96%.

Regarding CWW, the calves carrying the *Q204X* allele in the heterozygous form in the studied population gained 8.56 kg more weight than their counterparts not carrying the allele. From the point of view of the *F94L* allele, there was a difference of 4.08 kg in favor of the heterozygous individuals, but this difference could not be verified statistically. The weight of the individuals carrying the *nt324* and *nt414* alleles in the homozygous form was higher (10.43 kg and 2.92 kg, respectively) than the noncarriers, but these differences were not significant either.

Regarding the muscularity scores, it could be established that calves carrying the *Q204X* allele in the heterozygous form achieved approximately a 0.14 point higher MSB, MST and LTS, and a 1.2% higher OMP than their noncarrying partners. Despite the fact that the *F94L* allele had no statistically verifiable effect on muscularity parameters, it was striking that noncarrier calves showed higher values in almost all muscularity scores than heterozygous carriers. In the case of the *nt267* allele, the muscularity score of the heterozygous calves was higher—although not significantly—than that of the noncarrier individuals, and in the case of the *nt324* and *nt414* alleles even more so in the homozygous carriers.

In the case of all traits, we observed considerable differences between the individuals born in different years. This was also supported by the results of the phenotypic trend calculation (Table 6), according to which six of the nine examined traits were statistically reliable (*p* < 0.05 and *p* < 0.01), and fairly well matched (R^2^ = 0.57−0.93) regression functions were obtained. In the case of BIW and CWW, the slope of the straight lines (b) was in a positive increasing direction, while in the case of the other traits it was in a negative decreasing direction. Here it must be noted that, in the case of muscularity parameters, the annual decrease is very small, typically −0.05 or −0.07 points/year.

Based on the obtained phenotypic correlation values (Table 7), it could be established that the calving and weaning traits did not show a close relationship with each other or with the muscularity traits (r = 0.00−0.24). On the other hand, there was a close (r = 0.61−0.92) and statistically reliable (*p* < 0.01) correlation between the muscularity scores.

## 4. Discussion

The *myostatin* gene (*MSTN*) or sometimes called growth and differentiation factor 8 (*GDF8*) is a major negative regulator of skeletal muscle mass and differentiation, but *MSTN* also exists in smooth muscles [26]. In addition to muscle tissue, the influence of the *MSTN* on bone development has been established [27]. Moreover, *MSTN* causes a variety of metabolic changes affecting glucose and lipid metabolism and total bile acid content [28], as well as resulting in changes in semen characteristics [29]. An association was observed between the mutation in *MSTN* and susceptibility to a skin disease [30].

It is well known that there are several mutations in the coding region that have been detected as disruptive mutations (deletions, insertions and nucleotide substitutions) and they are thought to inhibit the function of the *MSTN* protein and are strongly associated with the double-muscling phenotype [22,31].

The *F94L* allele, a missense variant, was characterized by the substitution of cytosine by adenine at the nucleotide position of 282 in exon 1, which led to causing substitutions of leucine (Leu) for phenylalanine (Phe) at the 94th amino acid in the *MSTN* gene. Interestingly, the *F94L* mutation was not considered to cause a loss of *MSTN* function, which led an intermediate muscling in Charolais cattle [22].

The *Q204X* allele of this gene is a disruptive variant, and heterozygous carriers in the Charolais population had a greater mean carcass weight and conformation estimated breeding values (EBVs) [32], but this allele also caused calving difficulties and fertility problems [33].

We are interested in further three silent mutations, i.e., the polymorphisms of the *myostatin* gene caused by the *nt267*, *nt324* and *nt414 MSTN* mutations.

In our study, during the evaluation of the effect of *myostatin* alleles, *Q204X* was statistically proved to have an effect on the 205-day weaning weight, muscle score of back, muscle score of tight, loin thickness score and muscle development percentage. Calves carrying the *Q204X* allele in the heterozygous form in the studied population were heavier than those not carrying this allele. However, animals carrying the *F94L* allele in the heterozygous form were also heavier, but the difference was not significant. The weaning weights of calves carrying the *nt324* and *nt414* alleles in the homozygous form were higher than the noncarriers, but these differences were not significant either.

Similar to the results of our work, several previous sources [8,21,34] contain information on the statistically verifiable effect of the *Q204X* allele on meat production related traits. Contrary to our results, several previous studies [6,16] found the effect of the F94L allele to be significant on some muscularity-related parameters. Among the alleles belonging to the “small” *myostatin* group, we only found information on the effect of the double-muscled related allele *nt821* in existing sources [31,35,36]; however, this allele did not have an effect in the tested Charolais stock. The genetic structure of the *nt267*, *nt324* and *nt414* alleles was previously described by Dunner et al. [21], but no literature data were found on their effect on the phenotypic results.

The results of our work are similar to the findings of Casas et al. [12], according to which *myostatin* alleles in heterozygous form can have a favorable effect on weaning traits. Contrary to the results of Allais et al. [8], we could not detect the effect of the *Q204X* allele on birth weight in the examined Charolais herd. Similar to the results of Esmailizadeh et al. [22], the effect of the *F94L* allele on birth and weaning traits was not found to be significant. Our results are in line with Zhao et al.’s [29] findings: the *MSTN*-gene-edited Chinese Yellow cattle had improved growth traits compared with wild-type counterparts; however, the birth weight yielded no significant difference among groups, but, with increasing month age, the weight gain rate of *MSTN*-gene-edited cattle was significantly higher.

In this study, the weaning weight of Charolais calves were similar to the data found in most of the relevant literary sources [37,38,39].

The *MSTN* polymorphisms have negative effects on their reproductive traits, for example, calving difficulties (dystocia) [27]. First, Arthur et al. [40] studied Charolais cross animals and reported a higher incidence of dystocia, which was associated with phenotypically muscular calves. Moreover, the height, width and area of pelvic opening in homozygous dams were significantly smaller compared with normal dams. As previously established [41], Charolais heterozygous calves were slightly heavier at birth, with no association with calving ease.

On the basis of the calving ease score observed during our work, it seems that there was fewer difficult calving in the studied herd than what was found in the literature [42,43] in the case of the Charolais breed. It can be explained by the fact that our calves, heterozygous for the double-muscle gene, are superior to normal cattle in terms of meat production traits and do not have calving problems.

We found very little information available in the literature about the conformation of Charolais calves related to their muscularity. Arango et al. [44] and Vallée et al. [45] published data on purebred and crossbred Charolais herds, but, due to the different methodology, we did not have the opportunity to compare them with our results.

A better muscular conformation in heterozygote (*E291X* variant) carcasses of Marchigiana bulls [10] reflected our statement about the muscularity score of the heterozygous calves in the cases of the *nt267*, *nt324* and *nt414* alleles. As previously stated by Ceccobelli et al. [10], the greater muscularity of heterozygous animals compared with normal ones could be a starting point to improving productive efficiency in beef cattle. Regarding these *MSTN* gene polymorphisms in Charolais cattle, to our knowledge, no such data exist in the literature.

It seems that *MSTN* calves had significant improvement in muscularity traits, as previously described in *MSTN*-gene-edited cattle [29]. Recently, Gaina and Amalo [46] found two SNPs (*c424* and *c467*) of the *MSTN* gene in (Bos indicus) cattle, which are associated with phenotypes of wither height, heart girth and hip height, but not with body weight or body length.

The differences by birth year and sex of calves in weaning weight obtained during our work are very well known in the literature [47,48]. However, we did not find any data for this kind of evaluation of the muscularity parameters of Charolais calves.

Similar to our results, Gutiérrez et al. [49] and Chud et al. [50] did not find a close correlation between BIW, CAE and CWW traits in the case of the Asturiana de los Valles breed of cattle, and in the case of the Nellore breed.

## 5. Conclusions

Since *Q204X* had the greatest effect on calving, weaning and muscularity-related traits, we think it would be advisable to pay attention to this allele in the breeding strategy, to increase the proportion of carriers from generation to generation. It would be advisable to repeat this test periodically, because, based on literature data too, it seems that the allele in its homozygous form could cause calving difficulties.

Based on the results, the favorable effect of the *F94L* allele was not detectable in our study, contrary to some literary reports, which could be a consequence of the proportion of animals carrying the allele (about 5.5%) being very small in the studied population. On the other hand, based on previous studies, the better phenotypic performance of individuals carrying the allele was more evident in the fattening and slaughter traits.

The proportion of calves carrying the *nt324* and *nt414* alleles was quite high (21.5% and 48.1%, respectively) in the examined Charolais population. However, in the literature, there was very little information about their effect on phenotypic performance. Based on our results, it seems that homozygous carrier individuals may have better growth performance-related traits than noncarrier individuals. Therefore, it would be advisable to pay more attention to this allele.

## Figures and Tables

**Table 1 animals-13-01895-t001:** Occurrence of *myostatin* alleles in the examined population.

Myostatin Allele	Genotype	Male Calves	Female Calves	Total
		Number of Animals		
*F94L*	Noncarrier	651	1282	1933
	Heterozygous	37	76	113
	Homozygous	0	0	0
*Q204X*	Noncarrier	606	1185	1791
	Heterozygous	82	173	255
	Homozygous	0	0	0
*nt267*	Noncarrier	633	1318	1981
	Heterozygous	25	40	65
	Homozygous	0	0	0
*nt324*	Noncarrier	547	1060	1607
	Heterozygous	132	277	409
	Homozygous	9	21	30
*nt414*	Noncarrier	357	705	1062
	Heterozygous	277	548	825
	Homozygous	54	105	159
Total		688	1358	2046

**Table 2 animals-13-01895-t002:** Basic statistics of the examined traits (number of animals for each trait 2046).

Trait	Mean	SD	CV%	Min	Max	Norm *	Hom #
BIW (kg)	43.63	5.99	13.74	21	70	0.07	0.11
CAE (score)	1.16	0.45	38.55	1	3	0.51	0.00
CWW (kg)	258.15	44.30	17.16	125	404	0.03	0.00
MSS (score)	5.54	1.10	19.91	2	9	0.18	0.06
MSB (score)	5.13	1.05	20.39	2	8	0.19	0.02
MST (score)	5.36	1.16	21.71	2	10	0.17	0.27
RST (score)	5.35	1.12	21.01	2	9	0.18	0.33
LTS (score)	5.26	1.07	20.45	2	9	0.18	0.13
OMP (%)	53.15	9.62	18.10	20	87	0.05	0.04

BIW = birth weight; CAE = calving ease; CWW = 205-day weaning weight; MSS = muscle score of shoulder; MSB = muscle score of back; MST = muscle score of thigh; RST = roundness score of thigh; LTS = loin thickness score; OMP = overall muscle development percentage. * Normality test: if *p* > 0.05, the normal distribution is confirmed; # homogeneity test: if *p* > 0.05, the homogeneity is confirmed.

**Table 3 animals-13-01895-t003:** Effect of the examined factors on the calving, weaning and the muscularity traits.

Factors	Traits								
	BIW	CAE	CWW	MSS	MSB	MST	RST	LTS	OMP
	*p*								
Birth year of calves	<0.01	<0.01	<0.01	<0.01	<0.01	<0.01	<0.01	<0.01	<0.01
Sex of calves	<0.01	<0.01	<0.01	<0.01	<0.01	<0.01	<0.01	<0.01	<0.01
*F94L*	NS	NS	NS	NS	NS	NS	NS	NS	NS
*Q204X*	NS	NS	<0.01	NS	<0.05	<0.05	NS	<0.05	<0.05
*nt267*	NS	NS	NS	NS	NS	NS	NS	NS	NS
*nt324*	NS	NS	NS	NS	NS	NS	NS	NS	NS
*nt414*	NS	NS	NS	NS	NS	NS	NS	NS	NS
Factors	The ratio of the examined factors in phenotype (%)								
Birth year of calves	8.53	19.19	3.84	1.95	1.26	1.52	6.63	2.44	1.97
Sex of calves	90.37	62.27	87.90	96.18	96.74	94.43	92.32	95.53	96.49
*F94L*	0.24	1.39	0.68	0.12	0.39	0.01	0.21	0.50	0.24
*Q204X*	0.00	2.05	5.29	0.63	1.04	1.79	0.04	1.07	0.81
*nt267*	0.01	6.12	0.03	0.10	0.03	1.30	0.07	0.10	0.16
*nt324*	0.07	1.02	1.17	0.35	0.06	0.24	0.07	0.08	0.03
*nt414*	0.16	4.54	0.38	0.40	0.21	0.19	0.28	0.03	0.07
Error	0.62	3.42	0.71	0.27	0.27	0.52	0.38	0.25	0.23
Total	100.0	100.0	100.0	100.0	100.0	100.0	100.0	100.0	100.0

BIW = birth weight; CAE = calving ease; CWW = 205-day weaning weight; MSS = muscle score of shoulder; MSB = muscle score of back; MST = muscle score of thigh; RST = roundness score of thigh; LTS = loin thickness score; OMP = overall muscle development percentage.

**Table 4 animals-13-01895-t004:** The effect of different factors on the calving and weaning traits.

Factors	N	Calving and Weaning Traits		
		BIW (kg)	CAE (Score)	CWW (kg)
Adjusted overall mean (±SE)	2046	43.65 ± 0.63	1.12 ± 0.05	269.07 ± 4.73
		Deviation from the overall mean		
Birth year of calves				
– 2015	195	−0.98	+0.16	−6.02
– 2016	51	−0.37	−0.10	−9.20
– 2017	139	−2.36	−0.02	−4.12
– 2018	296	+0.46	+0.00	−2.01
– 2019	540	−0.06	+0.04	+4.67
– 2020	597	+0.76	-0.02	+6.93
– 2021	228	+2.54	−0.05	+9.74
Sex of calves				
– male	688	+1.67	+0.05	+11.54
– female	1358	−1.67	−0.05	−11.54
*F94L*				
–noncarrier	1933	+0.17	+0.01	−2.04
– heterozygous	113	−0.17	−0.01	+2.04
*Q204X*				
– noncarrier	1791	−0.01	−0.01	−4.28
– heterozygous	255	+0.01	+0.01	+4.28
*nt267*				
– noncarrier	1981	−0.05	+0.04	−0.54
– heterozygous	65	+0.05	−0.04	+0.54
*nt324*				
– noncarrier	1607	+0.08	+0.00	−4.58
– heterozygous	409	−0.07	−0.02	−1.27
– homozygous	30	+0.00	+0.02	+5.85
*nt414*				
– noncarrier	1062	+0.13	+0.02	−0.67
– heterozygous	825	+0.11	+0.02	−1.57
– homozygous	159	−0.24	−0.04	+2.25

BIW = birth weight; CAE = calving ease; CWW = 205-day weaning weight.

**Table 5 animals-13-01895-t005:** The effect of different factors on the muscularity traits.

Factors	N	Muscularity Traits					
		MSS (Score)	MSB (Score)	MST (Score)	RST (Score)	LTS (Score)	OMP (%)
Adjusted overall mean (±SE)	2046	5.90 ± 0.11	5.39 ± 0.11	5.65 ± 0.12	5.54 ± 0.12	5.52 ± 0.11	55.86 ± 0.96
		Deviation from the overall mean					
Birth year of calves							
– 2015	195	+0.13	+0.15	+0.04	−0.19	+0.12	+0.61
– 2016	51	+0.06	+0.14	-0.25	+0.00	+0.20	+0.57
– 2017	139	+0.19	+0.06	+0.06	+0.41	+0.07	+1.44
– 2018	296	+0.09	+0.01	+0.22	+0.36	+0.08	+1.40
– 2019	540	−0.02	-0.03	-0.04	+0.03	+0.03	+0.00
– 2020	597	−0.21	−0.18	−0.05	−0.24	−0.28	−2.06
– 2021	228	−0.24	−0.16	+0.02	−0.36	−0.22	−1.97
Sex of calves							
– male	688	+0.47	+0.44	+0.35	+0.41	+0.47	+4.34
– female	1358	−0.47	−0.44	−0.35	−0.41	−0.47	−4.34
*F94L*							
– noncarrier	1933	+0.03	+0.06	−0.01	+0.04	+0.07	+0.43
– heterozygous	113	−0.03	−0.06	+0.01	−0.04	−0.07	−0.43
*Q204X*							
– noncarrier	1791	−0.06	−0.07	−0.07	−0.01	−0.07	−0.60
– heterozygous	255	+0.06	+0.07	+0.07	+0.01	+0.07	+0.60
*nt267*							
– noncarrier	1981	−0.04	−0.02	−0.11	−0.03	−0.04	−0.46
– heterozygous	65	+0.04	+0.02	+0.11	+0.03	+0.04	+0.46
*nt324*							
– noncarrier	1607	−0.11	−0.04	+0.00	+0.02	+0.00	−0.23
– heterozygous	409	−0.07	−0.01	−0.06	+0.05	−0.04	−0.29
– homozygous	30	+0.17	+0.05	+0.06	−0.06	+0.04	+0.52
*nt414*							
– non carrier	1062	−0.06	−0.03	−0.03	−0.05	+0.00	−0.27
– heterozygous	825	+0.03	+0.03	−0.03	−0.01	−0.02	−0.01
– homozygous	159	+0.03	+0.00	+0.05	+0.06	+0.01	+0.28

MSS = muscle score of shoulder; MSB = muscle score of back; MST = muscle score of thigh; RST = roundness score of thigh; LTS = loin thickness score; OMP = overall muscle development percentage.

**Table 6 animals-13-01895-t006:** The phenotypic trend of the estimated traits.

Traits	Slope (bX)			Intercept (a)			Fitting	
	b	SE	*p*	a	SE	*p*	R^2^	*p*
BIW (kg)	+0.54	0.20	<0.05	−1042.52	4407.67	<0.05	0.59	<0.05
CAE (score)	−0.01	0.02	NS	29.82	31.44	NS	0.14	NS
CWW (kg)	+3.18	0.44	<0.01	−6146.81	885.23	<0.01	0.91	<0.01
MSS (score)	−0.06	0.02	<0.05	134.90	38.18	<0.05	0.70	<0.05
MSB (score)	−0.06	0.01	<0.01	122.69	14.19	<0.01	0.93	<0.01
MST (score)	+0.01	0.03	NS	−16.01	59.19	NS	0.03	NS
RST (score)	−0.05	0.06	NS	103.80	115.26	NS	0.13	NS
LTS (score)	−0.07	0.02	<0.05	150.65	36.77	<0.01	0.76	<0.05
OMP (%)	−0.51	0.20	<0.05	1077.82	401.49	<0.05	0.57	<0.05

BIW = birth weight; CAE = calving ease; CWW = 205-day weaning weight; MSS = muscle score of shoulder; MSB = muscle score of back; MST = muscle score of thigh; RST = roundness score of thigh; LTS = loin thickness score; OMP = overall muscle development percentage.

**Table 7 animals-13-01895-t007:** Phenotypic correlation values between the estimated traits.

r	CAE	CWW	MSS	MSB	MST	RST	LTS	OMP
BIW	* 0.13	* 0.24	* 0.13	* 0.15	* 0.08	* 0.13	* 0.13	* 0.14
CAE		0.00	* 0.09	* 0.09	0.04	* 0.08	* 0.09	* 0.09
CWW			* 0.21	* 0.20	* 0.17	* 0.24	* 0.21	* 0.24
MSS				* 0.86	* 0.61	* 0.68	* 0.80	* 0.90
MSB					* 0.63	* 0.66	* 0.82	* 0.91
MST						* 0.67	* 0.62	* 0.79
RST							* 0.65	* 0.82
LTS								* 0.92

* *p* < 0.01; BIW = birth weight; CAE = calving ease; CWW = 205-day weaning weight; MSS = muscle score of shoulder; MSB = muscle score of back; MST = muscle score of thigh; RST = roundness score of thigh; LTS = loin thickness score; OMP = overall muscle development percentage.

## Data Availability

The data presented in this study are available on request from the National Association of Hungarian Charolais Cattle Breeders.

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
