# Peer review of "Relationship between Some Myostatin Variants and Meat Production Related Calving, Weaning and Muscularity Traits in Charolais Cattle"

_animals, 2023, doi:10.3390/ani13121895_

Round 1

Reviewer 1 Report

In this study F94L, Q204X, nt267, nt324 and nt414 alleles of the myostatin gene were examined in relation to BIW, CAE,CWW, MSS,MSB, MST,RST, LTS, and OMP of Charolais weaned calves in Hungary.

However, I do have some issues with the study as presented here in:

1. Please check if the relevant references are fully and correctly cited.

2. This article has too little content, and only simple statistics cannot display the high-quality level of the article.

3. Whether the correct statistical method for analysis?

No major issues found, there are a few minor issues that need to be addressed.

Author Response

Cover letter

Manuscript ID: animals-2405482
Type of manuscript: Article
Title: Relationship between some myostatin variants and meat production
related calving, weaning and muscularity traits in Charolais cattle
Authors: Tamás Csürhés, Ferenc Szabó *, Gabriella Holló *, Edit Mikó,
Márton Török, Szabolcs Albin Bene

Dear Editors and Reviewers,

The authors are grateful for the tiresome work and time devoted to reviewing the manuscript, as well as for the comments and useful suggestions.

We try to respond point by point to the reviewers' comments and suggestions.

Reviewer 1. 

Comments: In this study F94L, Q204X, nt267, nt324 and nt414 alleles of the myostatin gene were examined in relation to BIW, CAE,CWW, MSS,MSB, MST,RST, LTS, and OMP of Charolais weaned calves in Hungary.

However, I do have some issues with the study as presented here in:

  1. Please check if the relevant references are fully and correctly cited.
  2. This article has too little content, and only simple statistics cannot display the high-quality level of the article.
  3. Whether the correct statistical method for analysis?

Responds:

Years ago, the National Association of Hungarian Charolais Cattle Breeders (17)  asked a scientist (18) to compile a list of myostatin variants that should be tested in the breed. Since then, these five variants have been included in the association's regulations, which are being investigated. The mentioned weight and muscling traits are in the Guideline of National Association of Hungarian Charolais Cattle Breeders (17). 

1.Relevant references were checked and corrected where it was necessary. Ten new literary sources were incorporated into the manuscript for improving the content.

2-3. The content was developed in the dicsussion and conclusion  chapters. Agreeing with the reviewer, the statistical method used is really simple. However, we believe that it is consistent with the purpose of the study. Based on the aim of the study, and considering the available data, the applied variance analysis, linear regression and correlation calculation gave reliable results. For this purpose, according to our opinion, complicated methods such as breeding estimation etc. were not necessary. Please clarify us, which methods should be used for the further evaulation.

The text has been checked and improved in several places.

Regards,

the Authors.

Reviewer 2 Report

Dear Authors,

The topic reported in your manuscript is interesing for whome are expert in the field.

The design of the study is valid and the adopted methods and the evaluated parameters are in line with the aim. It would be more promising if myostatin variants and markers analyzed and considered by you were more significant to apply in genetic cross-breeding strategy, for the improvement of bovine performance and quality of meat. Did you chose these variants well considering their eterogeneity and variability in the studied breed?

Anyway, my comments are below:

Introduction 

Lines 40-42: this sentence is not clear, please reformulate.

Line 47: please improve the significance of the sentence 

Lines 61-74: this part of the text is more suitable for discussion, 

so please valorize the introduction in a different way, for example focusing and dealing  with performance, quality and genetic traits of Hungarian Charolais.

Materials and Methods 

Line 91: please remove of 

Lines 98: please adjust the formatting.

Line 146: the sentence is incomplete.

Results 

Line 158: Also the birth year (independent variable) is statistically significant

Lines 186-188: please better explain this concept.

Discussion 

Lines 218-221: this period has to be improved and reformulated, to discuss your results.

As the same way, the entire discussion section is little weak.

References: please check the style and the format required by Animals Journal 

The quality of presentation could be improved.

Moderate english revision is necessary. 

The data reported in tables are clear and simple to understand.

Author Response

Cover letter

Manuscript ID: animals-2405482
Type of manuscript: Article
Title: Relationship between some myostatin variants and meat production
related calving, weaning and muscularity traits in Charolais cattle
Authors: Tamás Csürhés, Ferenc Szabó *, Gabriella Holló *, Edit Mikó,
Márton Török, Szabolcs Albin Bene

Dear Editors and Reviewers,

The authors are grateful for the tiresome work and time devoted to reviewing the manuscript, as well as for the comments and useful suggestions.

We try to respond point by point to the reviewers' comments and suggestions.

Reviewer 2.

Question.

Did you chose these variants well considering their heterogeneity and variability in the studied breed?

Reply:

Years ago, the National Association of Hungarian Charolais Cattle Breeders asked a scientist (18) to compile a list of myostatin variants that should be tested in the breed. Since then, these five variants have been included in the association's regulations, which are being investigated.

Comments- replies

Lines 40-42: this sentence is not clear, please reformulate.- Improved

Line 47: please improve the significance of the sentence – Improved

Lines 61-74: this part of the text is more suitable for discussion, - so please valorize the introduction in a different way, for example focusing and dealing  with performance, quality and genetic traits of Hungarian Charolais.

 Reply: Discussion is improved according to the suggestion.

Materials and Methods

Line 91: please remove of – Removed

Lines 98: please adjust the formatting. – Formated

Line 146: the sentence is incomplete. Improved  

Results

Line 158: Also the birth year (independent variable) is statistically significant – It was added into the sentence.

Lines 186-188: please better explain this concept. – Text is improved.

Discussion

Lines 218-221: this period has to be improved and reformulated, to discuss your results.

As the same way, the entire discussion section is little weak.

Reply: Discussion is improved according to the suggestion, and completed with 10 new literary sources.

References: please check the style and the format required by Animals Journal – Checked.

The text has been checked and improved in several places.

Regards,

the Authors.

Reviewer 3 Report

1. It is recommended to revise the conclusion section in the abstract, as the conclusion is not a list of results.

2. Please polish your English abstract.

3. Please write the gene names in italics throughout the text.

4. There are multiple instances of unclear sentences in the text. Please read and modify them thoroughly.

5. Some punctuation mark (half width and full width, etc.), font spacing, spaces, etc. in the full text need to be carefully checked and modified.

The English language quality of the entire article is not a major issue. Please carefully check whether the grammar and sentence structure meet the requirements of academic English

Author Response

Cover letter

Manuscript ID: animals-2405482
Type of manuscript: Article
Title: Relationship between some myostatin variants and meat production
related calving, weaning and muscularity traits in Charolais cattle
Authors: Tamás Csürhés, Ferenc Szabó *, Gabriella Holló *, Edit Mikó,
Márton Török, Szabolcs Albin Bene

Dear Editors and Reviewers,

The authors are grateful for the tiresome work and time devoted to reviewing the manuscript, as well as for the comments and useful suggestions.

We try to respond point by point to the reviewers' comments and suggestions.

Reviewer 3.

Comments and Suggestions:

  1. It is recommended to revise the conclusion section in the abstract, as the conclusion is not a list of results.
  2. Please polish your English abstract.
  3. Please write the gene names in italics throughout the text.
  4. There are multiple instances of unclear sentences in the text. Please read and modify them thoroughly.
  5. Some punctuation mark (half width and full width, etc.), font spacing, spaces, etc. in the full text need to be carefully checked and modified.

Comments on the Quality of English Language

The English language quality of the entire article is not a major issue. Please carefully check whether the grammar and sentence structure meet the requirements of academic English

Reply:

1.Conclusion and abstract section is improved according to suggestion.

  1. English abstracct is a little bit polished.
  2. Letters are changed.
  3. The sentences are improved.
  4. The full text is checked and modified where an inaccuracy were found.

The text has been checked and improved in several places.

Regards,

the Authors.

Round 2

Reviewer 1 Report

Nothing